# Silica Fouling in Reverse Osmosis Systems–*Operando* Small-Angle Neutron Scattering Studies

**DOI:** 10.3390/membranes11060413

**Published:** 2021-05-30

**Authors:** Vitaliy Pipich, Thomas Starc, Johan Buitenhuis, Roni Kasher, Winfried Petry, Yoram Oren, Dietmar Schwahn

**Affiliations:** 1Jülich Centre for Neutron Science JCNS-FRM II, Outstation at FRM II, Lichtenbergstr. 1, D-85747 Garching, Germany; v.pipich@fz-juelich.de; 2Neutron Scattering and Soft Matter (JCNS-1/IBI-8), Forschungszentrum Jülich GmbH, D-52425 Jülich, Germany; T.starc@fz-juelich.de; 3Biomacromolecular Systems and Processes (IBI-4), Forschungszentrum Jülich GmbH, D-52425 Jülich, Germany; j.buitenhuis@fz-juelich.de; 4Zuckerberg Institute for Water Research, Jacob Blaustein Institutes for Desert Research, Ben-Gurion University of the Negev, Midreshet Ben-Gurion 8499000, Israel; kasher@bgu.ac.il (R.K.); yoramo@bgu.ac.il (Y.O.); 5Heinz Maier-Leibnitz-Zentrum (MLZ), Technische Universität München, Lichtenbergstr. 1, D-85748 Garching, Germany; winfried.petry@frm2.tum.de

**Keywords:** silica fouling, cake formation and dissolution, reverse osmosis membranes, operando small-angle neutron scattering

## Abstract

We present operando small-angle neutron scattering (SANS) experiments on silica fouling at two reverse osmose (RO) membranes under almost realistic conditions of practiced RO desalination technique. To its realization, two cells were designed for pressure fields and tangential feed cross-flows up to 50 bar and 36 L/h, one cell equipped with the membrane and the other one as an empty cell to measure the feed solution in parallel far from the membrane. We studied several aqueous silica dispersions combining the parameters of colloidal radius, volume fraction, and ionic strength. A relevant result is the observation of Bragg diffraction as part of the SANS scattering pattern, representing a crystalline cake layer of simple cubic lattice structure. Other relevant parameters are silica colloidal size and volume fraction far from and above the membrane, as well as the lattice parameter of the silica cake layer, its volume fraction, thickness, and porosity in comparison with the corresponding permeate flux. The experiments show that the formation of cake layer depends to a large extent on colloidal size, ionic strength and cross-flow. Cake layer formation proved to be a reversible process, which could be dissolved at larger cross-flow. Only in one case we observed an irreversible cake layer formation showing the characteristics of an unstable phase transition. We likewise observed enhanced silica concentration and/or cake formation above the membrane, giving indication of a first order liquid–solid phase transformation.

## 1. Introduction

Desalinated inland brackish groundwater has become an important source of drinking water in arid and semi-arid regions and is increasingly used to resolve the worldwide water shortage. Reverse osmosis (RO) technology is the most widely used process for desalinating brackish water; however, a major limiting factor of this approach is silica scaling of the membrane, especially during recovery that exceeds 75%, a common practice in brackish water desalination [1,2,3]. Silica scaling may occur when the concentration of dissolved silica in either the bulk water or at the vicinity of the membrane surface surpasses its solubility limit, which depends on the pH, ionic strength, temperature, and the hardness of the treated water [4,5,6,7]. Importantly, once silica scale is formed on the membrane surface, it is extremely difficult to remove by common procedures such as acid wash [1,7]. Silica precipitation and scaling can be alleviated by using commercial antiscalants [8,9], or by reducing silica concentration in the feed water by alumina adsorbents [10], electrocoagulation [2], or softening through a coagulation [11]. However, such pretreatments increase operational costs and can induce organic fouling and biofouling and thus various alternative approaches have been tested to deal with silica scaling. For example, the presence of sodium alginate in the feed water reduces silica scaling during the desalination process [12].

The mechanism of silica fouling is not yet fully understood as this is a complex phenomenon depending on feed chemistry, permeate flux, and membrane surface topology, as well as on process parameters such as applied pressure and flow modes. Furthermore, there are several interaction mechanisms between the colloidal particles, as discussed in Section 3, as well as between the colloid and the membrane surface [13,14]. In addition, it was found that mineral scaling [15] and membrane surface functional groups [16,17] due to organic fouling affect the extent and properties of colloidal fouling on RO and nanofiltration (NF) membranes. The process of colloidal fouling appears mostly to be controlled by the competition of the permeation drag force due to permeation flux and the electric double layer repulsion potential between colloid and membrane [8,9,10]. Another issue of membrane fouling is membrane surface roughness effecting enhanced colloidal fouling [3,11,12].

The colloidal interaction potentials and forces are calculated in Section 3 on basis of the DLVO theory for the silica dispersions of this study [18,19,20]. It is known that aqueous silica dispersions are more stable than predicted from DLVO theory, indicating a larger repulsive interaction between silica colloids. This additional non-DLVO repulsive interaction appears as short range monotonic repulsive force effective in a range of less than 40 Å and is attributed to the presence of a ~10 Å thick gel-like layer of silanol and silicic acid groups grown at the surface of silica colloids when exposed to water [21].

Using specially designed equipment for operando studies allows us to follow the evolution of relevant parameters of the feed far from and on the surface of the membrane. So far, colloidal fouling has been mainly studied by permeate flux experiments. SANS provides direct insight into the fouling process on a microscopic length scale as it determines colloidal size and volume fraction of the feed dispersion, delivers thickness and porosity of the cake layer, and distinguishes between gas, liquid, and crystalline states of the silica in dispersions. In our study, we explore the fouling behavior of two polyamide RO thin film composite (TFC) membranes exposed to colloidal silica dispersions of different size, concentration, and ionic strength. It might be of particular interest that the lattice parameter, thickness, and porosity of the cake layer could be determined with good accuracy.

## 2. Experimental

In this section, we present the relevant parameters of the used SiO_2_ colloids, the corresponding silica feed dispersions, and the two commercial RO membranes followed by some details about the neutron diffractometer. The SANS-RO desalination equipment is described in detail in Appendix A.

### 2.1. Silica Synthesis and Characterization

Two types of silica particles were used in this study. The first one is the commercial LUDOX^®^ TMA which was ordered from Sigma-Aldrich. Its average particle radius of 116 Å was determined from Transmission Electron Microscopy (TEM). The second type of silica was synthesized in-house and is denoted as SJ29. These colloids have an average particle radius of 215 Å, as determined from TEM and depicted in Figure 1a. The corresponding SANS scattering pattern of both silica colloids are shown in Figure 1b and named as Silica-130Å and Silica-250Å.

The SJ29 particles were prepared in two steps. First, silica core particles were synthesized according to Stöber et al. [22] and then an additional layer of silica was grown onto these core particles by seeded growth following Giesche [23]. All Ludox TMA and SJ29 colloidal silica dispersions were redispersed in an ammonia/Cl- buffer with a pH between 9 and 10 and an ionic strength of 10 or 1.2 mM. This buffer contains only monovalent ions (main ions: NH4^+^ and Cl^−^). All chemicals used were of high purity and ordered from Sigma-Aldrich.

The size of the colloidal particles was determined by TEM, Dynamic Light Scattering (DLS), and SANS as compiled in Table 1 together with literature values of amorphous silica as well as the ions in solution. Silica concentration, ionic strength, and pH of samples used are given below in Table 2. Electrophoresis measurements were performed with a Malvern Zetasizer 2000 on a 1 g/L Silica-250Å dispersion of 10 mM ionic strength with the result of a zeta potential of −77 mV. This means that the silica particles are highly negatively charged in the ammonia buffer.

### 2.2. Instrumentation

The device for operando SANS experiments is described in Appendix A, further developed than the one we presented in [25]. The principle layout of the membrane cell (RO-MC) is depicted and explained in Figure A1. It allows the detection of characteristic parameters of scaling and fouling at three positions of the membrane in the course of RO desalination under the most realistic conditions. Figure A2 depicts the two pressure cells implemented at the sample position of the SANS diffractometer. One cell, the empty cell RO-EC, is without membrane and allows the measurement of the actual feed solution far from the membrane, whereas the other cell, the RO-MC cell, is equipped with membrane and spacer and is where the desalination process takes place. The scattering of the RO-MC cell must be corrected for scattering from membrane and spacer, which is rather strong due to the porosity of the membrane. Therefore, the scattering from membrane and spacer must be subtracted from the total scattering of the cell in order to obtain the scattering from the silica colloids alone for further analysis. A detailed analysis of several RO and NF membranes as well as of a standalone polyamide layer with SANS and Positron-Annihilation Lifetime Spectroscopy (PALS) is found in Refs. [26,27], respectively. The permeate flux and electric conductivity was measured in parallel to the SANS experiments. In this way, a relevant “engineering” parameter can be combined with the microscopic information from fouling of the silica dispersions.

The neutron experiments were performed at the SANS instrument KWS3 for “very small-angles” (VSANS) operating at the Heinz Maier-Leibnitz Zentrum (MLZ) in Garching, Germany [28], with the following settings: 12.8 Å neutron wavelength, 1.2 and 1.3 m sample to detector distance, and 0.7 × 0.5 cm^2^ beam area at the sample. These settings cover a range of scattering vector Q from about 10^−3^ to 0.03 Å^−1^, thereby being sensitive to scattering particles between several nm (10 Å) to about 0.1 μm. The absolute value of Q is determined according to Q=4π/λ sin(δ/2) from the scattering angle (δ) and wavelength (λ) of the neutrons. Its inverse value, 1/Q, corresponds to the dimension of objects mainly contributing scattered neutrons to this range. The relevant scattering laws needed for analysis are presented in Appendix B. The scattered neutron intensities are determined as macroscopic cross-section dΣ/dΩ(Q) in units of cm^−1^, i.e., scattering cross-section per unit cm^3^ of the sample determined from calibration with the primary beam.

The thickness of the sample is a relevant parameter needed for absolute calibration of the scattered intensity, which in our case is represented by the channel height determined by the space between sieve and sapphire neutron entrance window of the RO-MC cell (Figure A1). The scattering curves of this manuscript represent the scattering signal from the feed solution far from and above the membrane (concentration polarization) and from possible cake layer formation. The height of the channel between membrane surface and entrance sapphire window also determines the velocity of the feed across the membrane (cross-flow). The RO-MC cell has three windows for neutrons to pass (lower, middle, and upper), thereby allowing the determination of the concentration polarization averaged over the channel height as well as the fouling properties of the membrane at three positions of the membrane (Figure A2 and Figure A4).

It is noticed that the height of the feed channel at the three windows of the RO-MC cell changes differently with pressure as compiled in Table A1. These numbers were obtained from neutron transmission and measurement of the outer cell dimension with a micrometer screw. Figure 2 shows the relative change of channel height versus pressure, i.e., the space for the feed in front of the membrane at the three window positions normalized to the channel height at ambient pressure, i.e., to 0.101 cm and 0.086 cm in case of the BW30LE and RO98 membranes, respectively. Reasons of the different heights of the feed channel are (i) the larger thickness of the RO98 membrane (Table A1) and (ii) the applied pressure along the membrane. Information about the producers of the RO membranes can be found in the title of Table A1 and in Ref. [26]. It is obvious that pressure induces with 35% and 40%, the strongest increase in channel height in the middle part of the cell. Such change in height has strong influence on the feed velocity along the membrane, which must be considered when discussing its scaling and fouling behavior.

### 2.3. Experimental Procedure

Characteristic parameters of the examined aqueous silica dispersions are compiled in Table 2. The dispersions of the two silica colloids were prepared with nominal silica concentrations between 0.4 and 4.74 vol%, ionic strength of 1.2 and 10 mM, and pH between 9 and 10. Under these conditions, silica nanoparticles are negatively charged and thereby well stabilized [29]. The zeta potential as a measure of the charge near the surface of colloidal particles is obtained from the electrophoretic mobility. Following O’Brien and White ([30], Figures 3 and 4), the accuracy of the zeta potential is limited for dispersions with particle size and ionic strength of the present study (Table 2) as the electrophoretic mobility is showing a maximum at larger zeta potential for κR≅2–8 (κ Debye-Hückel parameter). Therefore, the zeta potential is ambiguous as the electrophoretic mobility is the measured quantity. Fortunately, this ambiguity can be resolved by comparing the zeta potential of dissolutions of much larger particles at about 9.5 pH and different ionic strength [31] with those of colloids of similar size as ours [32]. One control measurement was made on our SJ29 silica at 10 mM for which a zeta potential of −77 mV was obtained in reasonable agreement with literature [31,32]. For the 1.2 mM dispersion at about pH = 9.5, the relative change of the zeta potential with ionic strength was estimated from literature values [31,32] delivering −92 mV. 

The SANS-RO experiments were performed with the brackish water RO membrane BW30LE from Dow Filmtec and the seawater RO membrane RO98 pHt from Alfa Laval. The observations of this study are partly new and in some cases still need explanation. This is understood from the interplay of the applied parameters with the feed solution showing a complex interaction between the silica colloids as well as between colloid and membrane.

## 3. Theory: Stability of Aqueous Silica Dispersions (DLVO Theory)

We must consider two scenarios in our experiments. One is the bulk behavior of the colloidal silica dispersions and their stability against aggregation, the second is related to the interaction between colloids and membrane, with this second one largely determining the fouling characteristics. The first scenario is discussed in terms of the Derjaguin–Landau–Verwey–Overbeek (DLVO) theory [18,19,20].

Within the DLVO theory, the interaction potential between particles is described as a sum of a repulsive electric potential and an attractive van der Waals potential. The van der Waals attractive potential U_A_ in Equation (1) is valid in the case of (i) a much smaller distance between two spherical colloids (D) in comparison with the colloidal radius (R), i.e., D << R, and (ii) when obeying the non-retardation condition for approximately D < 100 Å [19]. The parameter A_H_ is the Hamaker constant, which for silica dispersions in aqueous solvents is calculated as $8.5\times 10^{-21}J$ [19]. An often-used description of the electrostatic repulsion potential is given in Equation (2), which has been derived within the linear superposition approximation [19], with $\varepsilon_{0}$ and $\varepsilon_{r}$ being the vacuum permittivity and the relative dielectric constant of the solvent, respectively, $\psi_{0}$ the Debye–Hückel surface potential, and 1/κ the Debye screening length of the medium. The surface potential $\psi_{0}$ is estimated from the zeta potential and the Debye length is calculated according to $\kappa =\sqrt{2000F^{2}I/{(\varepsilon }_{0}\varepsilon_{r}N_{A}k_{B}T)}$, where F is the Faraday constant, N_A_ the Avogadro number, k_B_ the Boltzmann constant, T the absolute temperature, and I the ionic strength in mol/L. The ionic strength of monovalent ions is given by $I\;=\;\sum_{i}c_{i}/2$ with c_i_ the concentration of ion i in mol/L. We evaluated an inverse Debye–Hückel parameter of 1/κ=87.8 Å and 30.4 Å for solutions of 1.2 and 10 mM ionic strength, respectively, whose interaction potentials are shown in Figure 3a. The interparticle force was evaluated from the interaction potential according to $F=-\;\partial U_{S}/\partial D$ as depicted in Figure 3b versus the colloid distance D. For a more detailed discussion of the DLVO theory, the reader is referred to two recognized introductions to the subject, [18,19].
(1)$$U_{A}\left(D\right)\;{=-\;A}_{H}\;R\;/12\;D$$
(2)$$U_{R}\left(D\right)\;=\;4\pi \;\varepsilon_{0}\varepsilon_{r}\;\frac{R^{2}}{D\;+\;2R}\;\psi_{0}^{2}\;e^{-\kappa D}$$

Considering the calculated DLVO potentials in Figure 3, an interesting study on the aggregation behavior of Ludox TMA (Silica-130Å) colloids must be mentioned also using an ammonia buffer of similar pH as ours [33]. In that study, the aggregation rate of silica colloids was fitted with the DLVO interaction potentials for a series of high salt concentrations, resulting in interaction potential curves that, after extrapolation to our lower salt concentrations, agree reasonably well with the present curves in Figure 3. We note, however, that considering the approximate character of Equation (2) and the zeta potential not being the surface potential, the agreement is better than expected, partly due to possible compensating errors. Nevertheless, the agreement of the potentials in Figure 3 with those of the aggregation study is a strong indication of a reasonably good estimate of the real interaction potentials.

As reported in literature, aqueous silica dispersions are more stable than predicted from DLVO theory. Reason of this enhanced stability is a non-DLVO interaction contribution presumably caused from the formation of a thin gel-like layer of ca. 10 Å thickness of silanol and silicic acid groups that grow on the silica surface in the presence of water [21]. It must be noted that this contribution in conjunction with DLVO interactions might prevent aggregation after short contact between the colloids. However, a contact of particles for longer time as in highly compressed layers might still lead to a sintering-type of reaction by the formation of siloxane bridges between the particles.

It is discussed and appears widely accepted in literature that colloidal fouling is controlled beside cross-flow by the interplay between permeation drag due to permeate flux and electric double layer repulsion between colloid and membrane surface [34,35]. The interaction of silica colloids and membrane surface is thoroughly discussed in the paper by Zhu and Elimelech on the basis of permeation experiments [13]. At the pH of our solutions, the surface of polyamide TFC RO membranes possesses a net negative charge with a zeta potential of about −10 mV ([14], Figure 11). The negative charge of both constituents results in a repulsive interaction between colloid and membrane. Therefore, colloid deposition and fouling on top of the membrane need a sufficiently large drag force caused from the rate of permeate flux (permeation drag), which must overcome the double-layer repulsion force between colloid and membrane. Another relevant parameter is surface roughness of the membrane showing strong influence on fouling behavior, as discussed in [13] and quite recently in [36]. We will not discuss this issue as we only present studies of RO membranes of expectedly similar surface roughness.

## 4. SANS Results and Interpretation

In this section, we present the SANS data of our RO desalination experiments performed with various aqueous silica dispersions at two RO membranes (Table 2) exposed to different pressures and cross-flows (CF). We applied pressure fields up to 25 bar and cross-flows from 0 (dead-end) to 36 L/h, the last one corresponding to an average feed velocity of v ≅ 20 cm/s (Table A1). We first present the results from the Silica-130Å colloids followed by those of Silica-250Å. Pressure is always mentioned as the applied pressure, as the osmotic pressure (π) is not known.

### 4.1. Aqueous Silica-130Å Dispersions of 1.2 mM Ionic Strength and Concentration of 1 vol% (BW30LE RO Membrane)

We start with the aqueous Silica-130Å dispersion of 1 vol% nominal concentration and 1.2 mM ionic strength exposed to the brackish water RO membrane BW30LE, the external dead-end condition. Under this condition, the feed shows liquid-like behavior verified by the interparticle interference peaks in Figure 4a,b, characteristic for dispersions of enhanced colloidal concentration of repulsive interaction. These data were fitted with the hard-sphere model (Equation (A5)) depicted as solid lines delivering at 2.29 h (Figure 4a) a colloidal volume fraction of Φ_HS_ = (8.1 ± 0.1) vol% corresponding to an average distance of the colloidal centers of about 580 Å, i.e., D ≅ 330 Å. The fit was performed setting a fixed radius of 126 Å for the silica. The corresponding volume fraction determined on basis of the formfactor (Equations (A3) and (A7)) is with Φ = (5.86 ± 0.08) vol%, about 25% smaller (Figure 4c). The subsequent cross-flow of 23 L/h applied at t = 2.9 h strongly reduces the scattering signal (Figure 4a). Figure 4b shows the SANS data after starting the dead-end condition at t = 3.37 h. At first, a strong increase in scattering is observed again, which after 8 h is continuously declining to smaller concentrations of 0.06 vol% at the end. Interference from liquid-like colloids is also observed for 9 h, showing a peak Q_m_ shifting from about 7.8 × 10^−2^ to 9.4 × 10^−2^ Å^−1^ before disappearing, i.e., then representing a gas-like colloidal dispersion. Figure 4c shows the colloidal concentration from all three positions of the RO-MC and RO-EC cells and Figure 4d the electric conductive (EC) and permeate flux (PF), respectively. The colloidal concentration was determined from the second moment (Q2, Equation (A7)). Similar values were derived from the ratio of dΣ/dΩ(0) and colloidal volume (V_P_) (Equation (A3)).

Figure 4c is split in two figures representing data before 4 h and after 6 h because the neutron beam shutter was closed in the meantime. During the first 1.5 h of operation at CF = 5 L/h, the colloid concentration linearly increases from about 0.5 to 1 vol% at all positions of the membrane. The following dead-end setting strongly increases the concentration in front of the membrane at all three positions, i.e., to about 1.8 vol% for middle and upper window and even 5.9 vol% for the lower window. The RO-EC cell shows with 1.2 vol% no abnormal increase in silica concentration. The following cross-flow of 23 L/h equalizes the concentration to about 1 vol% and 1.5 vol% at all positions of the RO-MC and RO-EC cells, respectively. At t = 3.3 h, we changed to dead-end condition again with the same reaction of an immediately strong increase in concentration, particularly in the lower part of the RO-MC cell, achieving its maximum value (Figure 4b) after about 8 h before exponentially declining to about 0.06 vol% after 16 h of operation (Figure 4c). The upper and middle parts of the RO-MC cell show much smaller concentrations whereas the RO-EC cell shows a constant value of (0.42 ± 0.03) vol%.

The electric conductivity (EC) and permeate flux (PF) were taken during the first 3.8 h. For the last 11 h of the experiment, we only can give the average flux. The EC mirrors the actual colloidal feed concentration in the RO-EC cell as its sensor is part of the feed circle (Figure A3). Its main task is to control the two valves of the pistons filled with feed and DI water for keeping a constant salt concentration in the circuit of the feed circulating through both cells (Figure A3). The conductivity is continuously increasing during the CF = 5 L/h operation in parallel to the colloid concentration in both cells. The constant EC is shown during the first dead-end period between t = 1.79 h and 2.65 h, as expected from zero cross-flow, whereas the following CF = 23 L/h let EC increase again. The permeate flux shows some oscillatory behavior mapping changes of colloid concentration averaged over the front of the membrane. The strong variation of colloid concentration at dead-end condition first induces a stronger decline in permeate flux followed by recovery due to reduced concentration polarization. Our device failed to register the permeate flux during this part of experiment. Thus, an average permeate flux of 0.44 L/(h m^2^ bar) could be evaluated from the total permeate volume of this part of experiment. We furthermore estimated a delay of about 13 min of the permeate flux data with respect to the EC data and therefore with the experimental settings. We explain the exponential decline in silica concentration from cooperative flowing down of the colloids forming the concentration polarization layer becoming first visible in the upper window and most clearly in the lower window at the end. Such cooperative flow is driven by gravitational force becoming large enough at sufficiently high colloidal concentration.

### 4.2. Aqueous Silica-130Å Dispersions of 10 mM Ionic Strength and Concentration of 1 vol% (RO Membrane BW30LE)

We present here experiments of two Silica-130Å dispersions of 10 mM ionic strength and nominal silica concentration of 1 vol% applied at the RO membrane BW30LE. The first experiment was performed in the same cell and membrane of the former section after carefully cleaning. Figure 5a shows a selection of SANS data from the lower RO-MC cell measured at the conditions of 20 bar/CF = 15 L/h and 25 bar/CF = 5 L/h. The data were fitted with the form factor of spheres (Equation (A4)). The corresponding parameters obtained for the RO-MC and RO-EC cells are depicted in Figure 5b,c delivering an average radius of R = (124 ± 3) Å and (128 ± 1.5) Å and a silica volume fraction approaching a volume fraction of about Φ = 1 vol% after about 13 h and (1.13 ± 0.05) vol% after 5 h, respectively, in good agreement with TEM, DLS, as well as the nominal colloidal concentration (Table 1 and Table 2). The volume fraction was determined from the ratio of dΣ/dΩ(0) and silica volume times scattering contrast, the last one compiled in Table 1.

The scattering data from the lower part of the membrane performed during the last part of experiment at 25 bar and dead-end condition are shown in Figure 6. Two distributions of the silica colloids become visible started just after stopping cross-flow, namely scattering from individual disordered silica colloids dissolved as gas-like (form factor) dispersion and from silica aggregated to a crystal, i.e., a cake layer. The fit (solid line) was performed with Equation (A6), representing an incoherent superposition (i.e., of intensity) of spherical form factor (Equation (A4)) and Gaussian distribution function describing phenomenologically the scattering from crystalline silica. The form factor (dashed-dotted line) delivers colloid radius as well as volume fraction of the silica in the RO-MC cell depicted in Figure 6c. Scattering of the cake layer at t = 40 min is shown in Figure 6a (open spheres fitted by the dashed line) and 6b as obtained after subtraction of the scattering from the individual silica (dashed-dotted line in Figure 6a). A distinct first order, i.e., [1,0,0] interference peak, is observed in accordance with the periodicity of the crystal, i.e., $\Lambda \;{=2\pi /Q}_{m}$ = (255 ± 2) Å in good consistence with the dimension of the silica colloid (Figure 6b). A higher order (Bragg) reflection also becomes visible, indicating a well-ordered crystal with a porosity of ε = (0.47 ± 0.02) in good agreement with a compact simple cubic crystal of ε = (1 − Φ_P_) = 0.48 (Table A2). A more detailed analysis confirms the simple cubic crystal structure; the broad peak at larger Q shows the [111] reflection whereas the [110] reflection at $Q=3.45\times 10^{-2}\;{Å}^{-1}$ is eliminated from the silica form factor, becoming zero at $Q=3.59\times 10^{-2}\;{Å}^{-1}$.

Figure 6c shows the silica volume fraction of feed and cake layer representing values averaged over the channel height of 0.109 cm between membrane (thickness of the membrane, 150 μm (Table A1)) and sapphire window (Figure A1). Figure 6d shows the evolution of the individual colloids in the middle and upper position of the membrane, showing an order of magnitude smaller volume fraction and no cake layer. The volume fractions of the silica are exponentially declining with time as well as the thickness of the cake layer from about 25 to 5 μm. It happens again that cake layer formation, particularly at the lower part of the membrane, is observed immediately after stopping cross-flow and its decline after 3 h in parallel with silica concentration by a factor of 4 for the same reason as for the strongly accumulated liquid-like colloids of the 1.2 mM Silica-130Å dispersion in Figure 4. Figure 6e shows the corresponding permeate flux slightly increasing with time.

Figure 7 presents SANS data from a second run of this experiment with a new RO-MC cell showing an unstable cake layer formation at the lower part of the membrane. This experiment was always performed at 25 bar. During the cross-flow of CF = 18 and 10 L/h, we observe individual (gas-like) Silica-130Å colloids of (126.6 ± 0.6) Å radius and (1.69 ± 0.3) vol% concentration in quite consistence with the results from the RO-EC cell (not shown). However, after changing to CF = 5 L/h, a sudden dramatic increase in silica concentration happens in the feed above the lower part of the membrane with the result of an irreversibly formation of a cake layer finally filling the whole channel. The scattering pattern in Figure 7a,b shows the corresponding change of scattering intensity and its shape. During the first 2 h (Figure 7a,c), we observe a strong increase in scattering driven by the silica volume fraction to a value of 10 vol% and the characteristic strong interparticle interference effect, showing at t = 25.3 h some intermediate shape from individual liquid-like colloids and crystallized ones in a cake (Figure 7b) before becoming stable at 26.4 h. No further change of scattering occurs. The periodicity (lattice parameter) of the cake is determined with Λ = (243 ± 0.3) Å, about 5% smaller than the diameter of the Silica-130Å colloids of 2R = (253 ± 1) Å, indicating a cake volume fraction and porosity of Φ_P_ = (0.59 ± 0.01) and ε = (0.41 ± 0.01). The slightly larger compactness or smaller porosity of the cake in comparison with a simple cubic lattice indicates a sintering-type reaction, the possibility of which was already mentioned in Section 3. The reason for this cake formation appears unclear to us and might be initiated by an accidental heterogeneous nucleation event as no abnormal behavior is observed at the upper parts of the membrane and the RO-EC cell.

### 4.3. Aqueous Silica-130Å Dispersions of 10 mM Ionic Strength and Nominal Concentration of 4.74 vol% (RO98 pHt-Alfa Laval Membrane)

In this section, we again present experimental data of an aqueous Silica-130 Å dispersion of 10 mM ionic strength but with a larger nominal silica concentration of 4.74 vol% and the RO98 pHt membrane. Figure 8a,b show SANS data of the silica dispersion operated at 25 bar and at varying cross-flow between 10 and 34 L/h (see Figure 8c). The scattering patterns of the RO-EC and RO-MC cells show an interparticle interference peak at about $Q_{m}=10^{-2}\;$Å^−1^ characteristic for liquid-like colloidal dispersions of larger concentration stabilized by the repulsive double layer interaction. The fit of the scattering pattern in Figure 8b is not so good seemingly from revealing inhomogeneities of the colloidal dispersion in front of the membrane due to concentration polarization. Such inhomogeneities are not observed for the RO-EC data from the much better fit. Figure A6b also shows the form factor measured for the RO-EC cell at t = 20.6 h.

The silica concentration is relatively constant in time during desalination and homogeneous at the positions of the membrane (Figure 8c) despite the great differences in cross-flow and large concentration polarization indicated by the nearly twice as large number. No cake layer formation is observed! Figure 8d shows a strong decline in the corresponding permeate flux (PF) following the usual exponential decline.

### 4.4. Silica-250Å Colloids of 10 mM Ionic Strength and Nominal Volume Fraction of 0.4 vol% (BW30LE Membrane)

In this section, we present SANS data of the larger Silica-250Å colloid dispersion of 10 mM ionic strength and nominal volume fraction of 0.4 vol%, which was exposed to the BW30LE membrane at conditions of 15 bar and 25 L/h. Figure 9 shows the permeate flux as well as SANS results from the RO-EC cell. The experiment was started with compacting the membrane by applying DI-H_2_O for about 18 h at 20 bar and 12 L/h cross-flow. An exponential decline (solid curve) of the permeate flux is observed from about 1.2 to 0.76 L/(h m^2^ bar) in Figure 9a, which is equivalent to the increase in hydraulic membrane resistance $R_{m}$, i.e., the inverse ratio of permeate flux and applied pressure (J(t)/P), from 0.83 to 1.32 in units of (h m^2^ bar/L). This permeate flux is smaller between a factor of 4 and 6.3 than the value of (4.78 ± 0.26) L/(h m^2^ bar) measured in the laboratory of one of the coauthors (see discussion in Appendix C). This means that the permeation drag force caused from permeate flux in our studies is much weaker compared with common RO desalination processes, thereby inducing a weaker concentration polarization and cake formation [24].

The permeate flux of the silica dispersion in Figure 9b shows a linear decline for about 5 h to 0.38 L/(h m^2^ bar) before a slightly linear increase to about 0.45 L/(h m^2^ bar) sets in. The change of the two scattering patterns of the RO-EC cell at 0.8 and 12.4 h in Figure 9c corresponds to the increase in colloidal volume fraction. Both scattering patterns were fitted with the form factor of spheres (Equation (A4)) depicted as solid lines. Figure 9d shows the corresponding colloidal volume fraction and radius dispersed in the feed present in the RO-EC cell far from the membrane; an average colloid radius of (243 ± 4) Å and nominal volume fraction of (0.42 ± 0.02) vol% was achieved after 4 h of operation. The smaller volume fraction at the beginning of the experiment occurred during filling the silica dispersion from dilution with rest of DI-H_2_O from the compaction experiment. There appears a correspondence between permeate flux and colloidal volume fraction; the permeate flux stops its decline after the colloids achieved their stable nominal concentration with a small delay of about 10 minutes. After 17 h of operation, the desalination process was stopped and a cleaning process with DI-H_2_O at 1 bar and 25 L/h cross-flow was started with the result of a finally strong decline in silica feed concentration ${(\Phi }_{\mathrm{feed}})$ to (0.06 ± 0.003) vol% as visualized by the red squares (∎) in Figure 9d.

Figure 10 depicts the SANS data from the middle part of the RO-MC cell, presenting time-dependent scattering with a pronounced increasing interference peak characteristic for the formation of a cake crystal. The scattering cross-section Δ{dΣ/dΩ}(Q) was obtained after subtraction of the scattering from the individual colloids as demonstrated in Figure 6a and Figure A6c and was afterwards fitted with the Gaussian function depicted as solid line. The SANS signals in Figure 10a are continuously increasing in amplitude and slightly shifting to larger Q, thereby indicating a gradual formation of a cake layer of increasing silica volume fraction and larger compaction, i.e., smaller porosity. Figure 10b shows the scattering pattern measured immediately after starting the 1 bar cleaning when the silica feed solution had not changed, as we know from the result of the RO-EC cell performed 10 min later (see first (∎) symbol in Figure 10d). The characteristic parameters of the cake layer are plotted in Figure 10c–e. The parameters A and Λ represent the amplitude and periodicity (lattice parameter) of the crystal (Figure 10c). The volume fraction of feed and cake (Figure 10d), as well as cake thickness (Figure 10e) are showing a continuous increase whereas the cake porosity (upper Figure 10e) is declining due to the shift of Q_m_ to larger Q (Figure 10a). After 16 h of operation, the periodicity (i.e., lattice parameter) achieved a value of Λ = (525 ± 5) Å and a constant volume fraction of (0.41 ± 0.04) vol% corresponding to a porosity of (0.59 ± 0.04) vol% (Figure 10e). At the same time, the cake thickness increases from 3 to 16 μm.

The feed volume fraction in the RO-MC cell achieves its nominal value only after 9 h, i.e., 5 h later than in the RO-EC cell, and there seems to be no concentration polarization (Figure 10d). Change to ambient pressure shortly enhances colloidal volume fraction in feed and cake. Cake porosity (ε) is determined from the ratio of $\varepsilon =\left[1-\;\left({4\pi R}^{3}/3\right){/\Lambda }^{3}\right]\;$ whereas the thickness of the cake ($\delta_{c}$) from $\delta_{c}=M_{d}\;/\left[d_{S}\left(1-\varepsilon \right)\right]$, with the mass M_d_ of cake per membrane surface and d_S_ the mass density of the silica (Table 1), or in our case, from the ratio of $\Phi_{\mathrm{cake}}/\left(1-\varepsilon \right)$ multiplied with the channel height achieving a maximum value of (16 ± 1) μm. Figure 11 depicts corresponding parameters from the upper and lower part of the RO-MC cell showing qualitatively the same behavior as the middle one.

The cleaning operation at ambient pressure shows some unexpected behavior indicated by the red squares (∎) in Figure 9, Figure 10 and Figure 11. At the first moment of cleaning, the cake layer remains stable, as shown in Figure 10b, with a slightly increased intensity and a shift to the smaller $Q_{m}\;=1.10\times 10^{-2}$ Å^−1^. Only after about 19 h (Figure 10c,d and Figure 11), i.e., 1 h after starting the cleaning procedure, the cake layer has disappeared in company with a strong decline in the individual colloidal concentration to a volume fraction of $\Phi_{P}$ = (0.034 ± 0.003) vol% in the whole system. Cleaning also leads to a 34% enhanced volume fraction of individual silica colloids before disappearing (Figure 10d and Figure 11), a behavior we do not observe in the RO-EC cell (Figure 9d).

The relaxation of the feed from 15 bar to ambient pressure caused a shift of peak position to smaller Q_m_, i.e., from $1.20\times 10^{-2}$ Å^−1^ to $1.10\times 10^{-2}$ Å^−1^ (Figure 10a,b), corresponding to a 9.1% larger periodicity Λ ((∎) in Figure 10c), i.e., an about 30% larger dilution of the crystal unit cell. As this measurement was performed with the original feed solution as mentioned above, the change of periodicity Λ, i.e., closer packing at higher pressure, is a measure of the compressibility of silica crystals in solution. From definition of the isothermal compressibility in Equation (3) with $V\;\equiv \;\Lambda^{3}$, one obtains $\kappa_{T}=-\;1.67\times 10^{-2}\;1/\mathrm{bar}$. This value is extremely large, if compared with the isothermal compressibility of water, which is $\kappa_{T}=4.6\times 10^{-5}\;1/\mathrm{bar}$ at 20 °C. However, considering the colloids of the cake layer behaving as an ideal gas in the feed dispersion, one derives with the help of the corresponding equation of state (P V=n RT) a $\kappa_{T}=1/P$ evaluating for 15 bar, four times larger compressibility of $6.7\times 10^{-2}\;1/\mathrm{bar}$. Isothermal cake compressibility in terms of colloidal volume fraction ($\Phi_{\mathrm{cake}}=N\times V_{P}$) with colloidal number density (N) and porosity (ε) of the cake is given in Equation (4). So far, cake layer porosity in membrane filtration was not directly available from the experiment, which is the reason to discuss cake layer compressibility in membrane scaling differently on the basis of a phenomenological model of transient hydraulic resistance R_c_(t) determined from the change of permeate flux due to cake formation as outlined below in Appendix C together with the results for the present experiment (Figure A7).
(3)$$\kappa_{T}\;=\;-\;\;d\;\left\{\ln\left(V\right)\right\}/\mathrm{dP}\mapsto \kappa_{T}\;=\;-\;\;3\;d\;\left\{\ln\left(\Lambda \right)\right\}/\mathrm{dP}$$
(4)$$\kappa_{T}\;=\;-\;\;d\;\left\{\ln\left(\Phi_{\mathrm{cake}}\right)\right\}/\mathrm{dP}=\;d\left\{\ln\left(1-\varepsilon \right)\right\}/\mathrm{dP}$$

### 4.5. Aqueous Silica-150Å Dispersion of 10 mM Ionic Strength and Nominal 1.09 vol% Volume Fraction (RO98 pHt-Alfa Laval Membrane)

In this section, we present SANS data from a dispersion of Silica-250Å colloids in a feed solution of 10 mM ionic strength and nominal volume fraction of 1.09 vol%, which was exposed to the RO98 pHt seawater membrane (Table 2). The membrane was compacted for 12 h before a pressure of 25 bar and three cross-flows of 18, 10, and 14 L/h were applied. Figure 12 shows the corresponding SANS data from the RO-EC cell. The two scattering patterns in Figure 12a were taken at the start and end of the desalination operation. We observe a slight influence of the structure factor S(Q) (Equation (A2)), i.e., a weak liquid-like behavior due to enhanced silica concentration, in combination with repulsive interactions between the silica colloids. Figure 12b shows the silica volume fraction and radius in the RO-EC during operation. The volume fraction was determined from ${d\Sigma /d\Omega (0)/V}_{P}$ and Q2 on the basis of Equations (A3) and (A7), respectively. The second moment Q2 delivers a 9% smaller value. After about 5 h, both SANS silica volume fractions achieved values between 0.84 and 0.92 vol%, slightly smaller than the nominal one, which again was attributed to dilution with rest DI-H_2_O from compaction during filling the feed into the circuit. The cross-flow shows no influence on the silica volume fraction.

Figure 13 shows the SANS data, i.e., Δ{dΣ/dΩ}(Q) from the cake layer formed at the three cross-flows in the middle part of the RO-MC cell. During the first 4 h before “start”, the pressure was continuously increased in 5-bar steps to 25 bar as visualized in Figure 13d,f from the increase in the concentration of the feed dispersion and increase in permeate flux. After the conditions of 25 bar and 18 L/h were set, the cake layer started to grow continuously, showing an increasing amplitude and slight compaction due to the shift of the interference peak (Q_m_) to larger Q (Figure 13a). This process was stopped and made undone by applying CF = 36 L/h at same pressure (cleaning), as scattering was only observed from individual colloids (Figure 13d,e). After setting CF = 10 L/h (Figure 13b) and = 14 L/h (Figure 13c), we always observe sudden cake formation, which afterwards is declining in amplitude just as observed for the Silica-130Å dispersions in Figure 4c and Figure 6c,d. For the CF = 14 L/h condition, the cake layer shows a relatively strong compaction (Figure 13c). Cleaning with 36 L/h cross-flow was performed between the transitions from 18 to 10 L/h and from 10 to 14 L/h cross-flow operations. Figure 13d shows volume fraction as well as radius (R) and half lattice parameter (Λ/2) of the individual and crystallized silica colloids, whereas Figure 13e shows cake porosity and thickness during operation. The decline in cake porosity below that of simple cubic crystals (Table A2) during the CF = 14 L/h condition was already observed for the process in Figure 7 and indicates partial sintering of the colloids to small solid blocks. A partial sintering process is also supported by approaching a constant thickness of about 1.6 μm after more than 40 h of operation at CF = 14 L/h.

Figure 13f shows the permeate flux of this operation. Again, formation of the cake layer at 18 L/h seems to support a slightly enhancing permeate flux, whereas its decline at 10 and in particular at 14 L/h leads to a slightly declining permeate flux. This observation might be attributed to the decline in cake porosity in particular because of partial sintering. The absence of cake layer and smaller colloid concentration during cleaning at 36 L/h are detected as small peaks of enhanced permeation. Figure 14 shows the corresponding colloidal parameters of the upper and lower part of the RO-MC cell. Cake layer formation of smaller volume fraction was observed at the two smaller CFs of 10 and 14 L/h.

A remark must be made for the radii of the colloids determined from the formfactor of spheres. In the RO-EC cell, we observed a constants radius of (245 ± 0.8) Å independent from cross-flow (Figure 12b), whereas we made the curious observation of a constantly smaller radius in the RO-MC cell in the case of cake formation. An average radius of (227 ± 1.2) Å were determined in the lower window and of (215 ± 3) Å in the middle and upper window. This means that a 5–10% smaller colloidal radius is determined above the cake layer, which in absence of the cake at CF of 36 L/h again becomes (240 ± 2) Å.

To prove the concept of spherical formfactor analysis we applied Guinier’s law, i.e., the form factor $F(Q)=\exp\left\{-{\left(R_{g}\;Q\right)}^{2}/3\right\}$, valid only in the small Q regime of roughly Q < 1/R_g_ with R_g_ the radius of gyration and its relationship to the radius of spheres according to $R=\sqrt{5/3}\;R_{g}$ [37]. This means that Guinier’s law is only valid in the small Q regime and thereby less influenced from scattering of the cake layer as the spherical formfactor analysis needs the whole Q. Figure 14c shows a Guinier plot, i.e., a plot of ln (F(Q)) versus Q^2^ of two scattering curves in the absence and presence of a cake layer. These curves were averaged over several scattering curves for achieving better statistics. The square root of the slope of the scattering curves in the presentation of Figure 14c determines R_g_. Different slopes of both data sets are clearly visible delivering a change of ${\Delta R}_{g}=\left(14\;\pm \;0.2\right)$Å or in radius of ΔR=(18 ± 0.3)Å in consistence with the results from the spherical formfactor analysis. The analysis with the formfactor of spheres in Figure 12a and Guinier’s law in Figure 14c appears to us as a good indication of the decline in silica colloids as a real effect in combination with our conclusions from partially silica sintering in Figure 13e. 

## 5. Discussion and Summary

We presented several examples of operando SANS experiments on silica fouling at two RO membranes under close to realistic conditions of RO desalination technique. The evolution of silica volume fraction far from and at three positions above the membrane, the lattice parameter of silica cake layer as well as its volume fraction, porosity, and thickness were determined and compared to simultaneous measured permeate flux. The detailed information of the cake layer formation, especially its reversibility and irreversibility (Figure 10e), as well as its effect on the permeate flux (Figure 9b), could become relevant for improving the operation of real full-scale RO desalination plants. Certainly, these first observations still need further detailed and focused operando SANS experiments.

The examples show that the formation of cake layer depends to a large extent on colloidal size, concentration, ionic strength, as well as cross-flow. It is shown that cake layer formation is basically a reversible process, which were nearly dissolved in two experiments with the feed at larger cross-flow of CF = 36 L/h (Figure 9 and Figure 10) and with DI-H_2_O at ambient pressure and CF = 25 L/h (Figure 13). Only in one case we observed a complete irreversible cake layer formation for the Silica-130Å dispersion (Figure 7). A schematic cross-section of the channel in the RO-MC cell is shown in Figure 15 with the dimensions of the middle window at 25 bar showing a cake layer as well as dispersed silica colloids on top of the BW30LE membrane. The orange curve qualitatively represents the exponential decay of concentration polarization in front of membranes. A cross-flow of 18 L/h corresponds to a cross-flow velocity of v ≅ 8.2 cm/s as evaluated from the cross-section (4.5 cm channel width) of 0.61 cm^2^ for channel and membrane. The upper and lower windows show a 78% smaller channel height, thereby a correspondingly larger cross-flow velocity of v ≅ 10 cm/s, which must be considered for the various pressure fields as depicted in Figure 2 and compiled in Table A1. At these conditions, laminar flow is dominating as a Reynold number of Re = 216 < critical Re (2300) is evaluated.

We first give a brief summary of the experimental results:Silica-130Å dispersion in Figure 4 (1 vol%, 1.2 mM, BW30LE membrane): Gas-like silica dispersion, no cake formation, and negligible concentration polarization at finite cross-flow of 5 and 23 L/h at 25 bar. Change to liquid-like dispersion (scattering shows interparticle interference peak) at 25 bar and dead-end condition due to large increase in concentration polarization but no cake formation.Silica-130Å dispersion in Figure 5 and Figure 6 (1 vol%, 10 mM, BW30LE membrane): Gas-like silica dispersion showing no cake formation or concentration polarization at finite cross-flow of 15 and 5 L/h at, respectively, 20 and 25 bar. Cake layer but no concentration polarization is formed at 25 bar and dead-end condition. The cake layer disappears for the most part within 3 hours, first in the upper window followed by the middle and lower ones. This observation allowed us to interpret the cake disappearance as a cooperative particle flow driven by gravity.Silica-130Å dispersion in Figure 7 (1 vol%, 10 mM, BW30LE membrane): Gas-like silica dispersion with no cake layer formation and concentration polarization at 18 and 10 L/h cross-flow. Switching from 10 to 5 L/h cross-flow changes the feed at the lower part of the membrane to a liquid-like dispersion due to strong increase in concentration polarization, which transforms within 5 hours to an irreversible cake layer filling the whole channel. This process indicated a first order liquid–solid phase transition. Meanwhile, the feed in the RO-EC cell showed only a slight increase in silica concentration.Silica-130Å dispersion in Figure 8 (4.74 vol%, 10 mM, RO98 pHt membrane): This liquid-like silica dispersion shows concentration polarization about twice as large as concentration averaged over the channel height in comparison with the feed in the RO-EC cell, and no cake layer formation. This experiment was always performed at finite cross-flow.Silica-250Å dispersion in Figure 9 and Figure 10 (0.4 vol%, 10 mM, BW30LE membrane): Gas-like silica dispersion, no concentration polarization at 15 bar and 25 L/h cross-flow, but temporal cake layer formation of increasing concentration and thickness and declining porosity. The nominal silica volume fraction of the feed in the RO-EC cell was achieved after about 4 h, whereas in the middle part of the membrane after 10 h. In parallel, the cake layer achieved a volume fraction of (0.56 ± 0.04) vol% (averaged over the channel height) after 14 h (Figure 10b). The upper and lower windows (Figure 11) show slightly smaller volume fractions of individual colloids and cake layer. Cleaning with DI-H_2_O at 1 bar and 25 L/h led the cake layer to re-dissolve after some delay. The cake porosity could be determined in the same feed at 25 bar and 1 bar, which allowed us to determine the cake compressibility. The fouling process appeared rather homogeneous at all three positions of the membrane.Silica-250Å dispersion in Figure 12 and Figure 13 (1.09 vol%, 10 mM, RO98 pHt membrane): The liquid-like dispersion shows concentration polarization as concluded from comparison of the volume fraction of (0.92 ± 0.01) vol% in the RO-EC cell with the over the process averaged feed concentration of (2.1 ± 0.2) vol% in the middle part of the RO-MC cell. Stable cake layer formation is observed for CF = 18 L/h but becomes unstable for the lower cross-flow of 10 and 14 L/h. During cake formation, we observed smaller silica sizes in the feed. Before changing cross-flow, the cake layer was completely removed after applying a cross-flow of 36 L/h. At CF = 14 L/h, the porosity became smaller than for simple cubic crystals, suggesting partial sintering of the silica colloids and also explaining the remaining cake thickness of about 1.6 μm. These observations indicate partially irreversible cake formation (Figure 13e).

### 5.1. Cake Layer Formation and Cleaning

Cake formation is less preferred for the smaller Silica-130Å colloids. Only the dispersion of 10 mM ionic strength and 1 vol% nominal colloid volume fraction showed cake formation at 25 bar and dead-end conditions at all three positions of the membrane (Figure 6). This cake layer formed spontaneously but became unstable by a collective particle flow driven by gravity. This mechanism might have been supported by the relatively low permeation drag force of the present experiments.

The larger Silica-250Å dispersions show a stronger tendency to cake layer formation as shown for both dispersions of 0.4 vol% and 1.09 vol%. The results of the 0.4 vol% dispersion seems particularly interesting and illuminating as it allowed a rather detailed analysis of the cake formation. Although concentration polarization was absent, a continuously increasing cake layer is observed, whose thickness, porosity, and cake resistance (R_c_) could be determined (Figure 10 and Figure A7). The 1.09 vol% dispersion exposed to 25 bar showed stable cake formation at 18 L/h always increasing in time, but cake dissolution at the smaller cross-flows of 10 and 14 L/h after fast formation (Figure 13). Two cleaning procedures at 1 bar and 25 L/h with DI-H_2_O (Figure 10 and Figure 11) as well as at 25 bar and 36 L/h with the silica dispersion (Figure 13d,e) removed the cake layer and is thereby approved as successful cleaning procedures of the membrane surface. These data clearly show that colloidal size as well as cross-flow and ionic strength play an essential role in cake layer formation. The effect of colloidal size on cake formation can be explained with the critical transmembrane pressure ${(\Delta p}_{C}=P\;-\;\pi )$ in Equation (5) that must be overcome for introducing cake layer formation ([38] p. 28, [39]). The parameter N_FC_ is the critical filtration number and the third power of R shows the strong influence of colloidal size on ${\Delta p}_{C}$.
(5)$${\Delta p}_{C}=\frac{3\;k_{B}T}{{4\pi R}^{3}}\;N_{\mathrm{FC}}$$

In some cases, the cake layer is forming very fast and afterwards disappearing to small volume fraction dependent of CF. The data indicate a cooperative flowing down of colloids of enhanced concentration due to concentration polarization or cake formation, when the interaction force between colloid layer and membrane (permeation drag) is not strong enough to carry their weight. Lower pressure (permeation drag) and larger cross-flow can support the process of dissolution. Furthermore, in most cases the cake layer was reversible and disappeared during cleaning at large cross-flow (Figure 10 and Figure 13d). This observation is in contrast to several statements in literature ([38] p. 28, [40,41]).

### 5.2. Change of Colloid Radius during Cake Formation

In case of cake formation of the Silica-250Å dispersion at the RO98 pHt membrane (Section 4.5), we surprisingly observed a decline in radius of about 13 Å for the silica colloids in the feed above the membrane (Figure 12, Figure 13 and Figure 14). To exclude an artefact from the combined analysis of free and cake-bound silica in Equation (A6), we repeated the analysis with Guinier’s law, as depicted in Figure 14c. This analysis gives the same result and better confidence as it is limited to the small Q regime of <1/R_g_ outside the range of significant scattering from the cake layer. SANS delivers a change of radius of ca. 18 Å, which is of similar size as the gel-like layer of silanol and silicic groups forming at the silica surface in the presence of water [21]. This gel-like layer effects a larger stability of silica dispersions due to a non-DLVO interaction [21]. It is not clear to us why this happens at all and only for Silica-250Å exposed to the RO98pHt membrane. This observation also stands out by large silica concentration polarization in front of cake layer and membrane. A loss of the gel-like layer at the silica surface would lead to a less stable dispersion and could be the reason of enhanced compaction of the cake by sintering (Section 3, Figure 13d).

### 5.3. Permeate Flux

During compaction of the BW30-LE membrane with DI-H_2_O at 20 bar and 12 L/h cross-flow, a permeate flux between 1.2 and 0.76 L/(h m^2^ bar) is found (Figure 9a). These values are smaller by a factor of 4–6.3 than the expected (4.78 ± 0.26) L/(h m^2^ bar) measured in the laboratory of one of the coauthors. However, several membrane pieces of the BW30-LE membrane also showed poor water permeability in the range of our data (0.74 to 0.96 L/(h m^2^ bar)). The lower permeate flux means a much weaker permeation drag force in comparison with common RO desalination experiments, thereby reducing the degree of concentration polarization and cake formation. This effect might be partly compensated by the larger colloid volume fraction in our experiments in the range of 1% in comparison with e.g., 0.008 vol% dispersion of 1000 Å silica particles in [24].

Quite generally, we must bear in mind that the permeation flux gives information on cake layer and concentration polarization averaged over the active membrane surface, which, according to the present SANS experiments, sometimes shows quite large inhomogeneous cake distribution. The experiment with 0.4 vol% Silica-250Å dispersion (Figure 9) in Section 4.4 might be a good example to compare the permeate flux with the corresponding SANS data, as we observe stable cake formation (Figure 10) distributed quite homogeneously along the membrane (Figure 11). A strong decline in permeate flux from about 0.8 to 0.4 L/(h m^2^ bar) is observed during the first 5 h (Figure 9b), which is accompanied by an increase in the colloidal concentration in the feed and a continuous cake formation (Figure 10d). Once the feed solution achieved its stable concentration in the RO-EC cell after 4 h no further decline in the permeate flux is observed, but instead a slight increase becomes evident even though the cake layer is further growing in thickness and declining in porosity. (Figure 10e). This means that a growing cake layer can become more permeable in contrast to predictions of Happel and Carmen-Kozeny ([11] Figure 2.7, p. 35). Furthermore, the discussion of cake layer porosity in Appendix C shows that the cake resistivity (R_c_) is much smaller than the hydraulic membrane resistivity (R_m_) (Equation (A8) and Figure A7). The maximal ratio of R_c_/R_m_ is of the order of 1%, showing that the cake layer has a negligible effect on permeate flux, which on the other hand means that permeate flux is essentially determined by the colloidal volume fraction in front of the membrane.

## Figures and Tables

**Figure 1 membranes-11-00413-f001:**
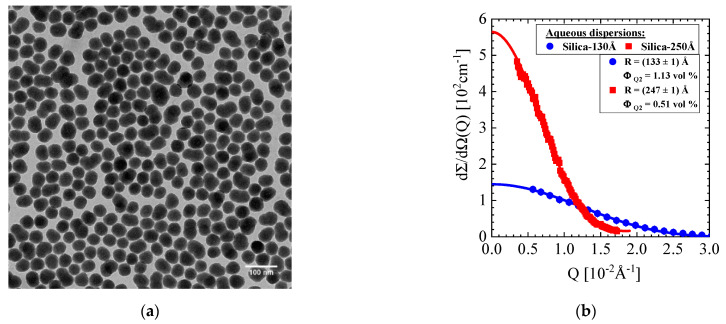
(**a**) TEM micrograph of SJ29 silica (Silica-250Å) and (**b**) SANS scattering curves of two silica colloid dispersions of 10 mM ionic strength. The SANS data of the Ludox (Silica-130Å) and SJ29 (Silica-250Å) were fitted with the form factor of spheres (Equation (A4)).

**Figure 2 membranes-11-00413-f002:**
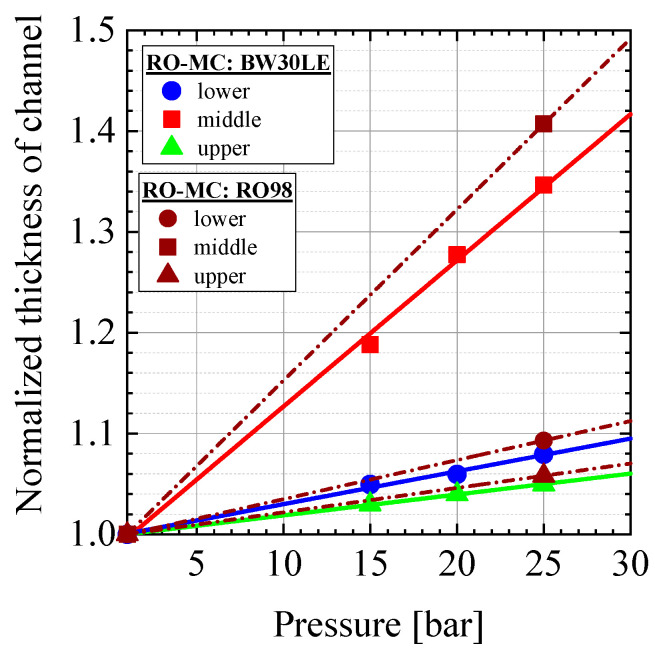
Change of channel height above the membrane versus pressure normalized to the height at 1 bar. These values were 0.101 and 0.086 cm for the experiments with the BW30LE and RO98 membranes, respectively. The values of the height are compiled in Table A1.

**Figure 3 membranes-11-00413-f003:**
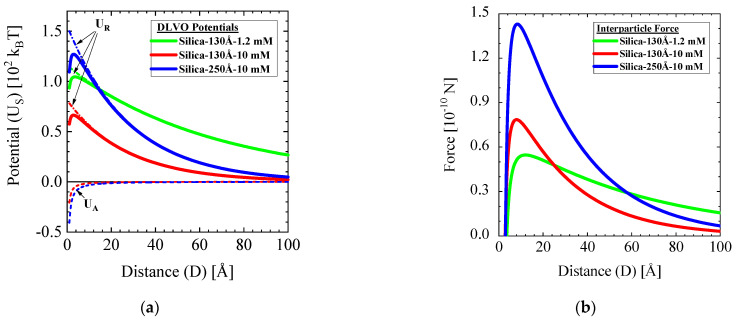
(**a**) DLVO interaction potentials in units of k_B_T of the silica solutions, i.e., the attractive (London force U_A_) the double layer repulsive (U_R_) and the sum of both interaction potentials (U_S_). (**b**) Force field between the silica colloids of the studied feed solutions.

**Figure 4 membranes-11-00413-f004:**
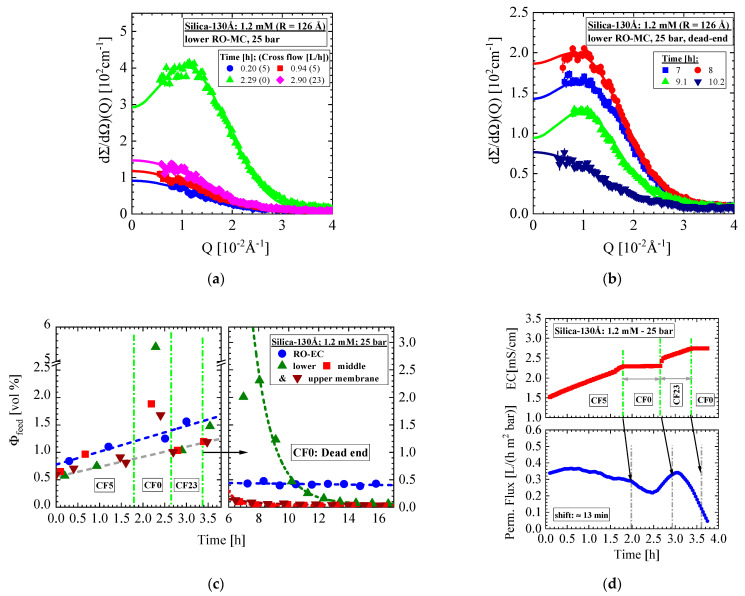
SANS results of aqueous Silica-130 Å dispersion of 1 mM ionic strength and 1 vol% nominal concentration exposed to 25 bar and various cross-flows (CF) from zero (dead-end) to 23 L/h. Unfortunately, we missed here the SANS data between 3.7 and 6 h of operation because of an unexpected closer of the shutter for neutrons. (**a**) Scattering data from the lower part of the of the RO-MC cell showing a strong increase in scattering with interference peak when stopping the CF to dead-end condition. (**b**) Scattering of the lower membrane position for various times of the second dead-end condition. (**c**) Silica concentration versus operation time of empty cell (RO-EC) as well the 3 windows of the RO-MC cell. (**d**) Permeate flux and electric conductivity during the first 3.5 h of operation.

**Figure 5 membranes-11-00413-f005:**
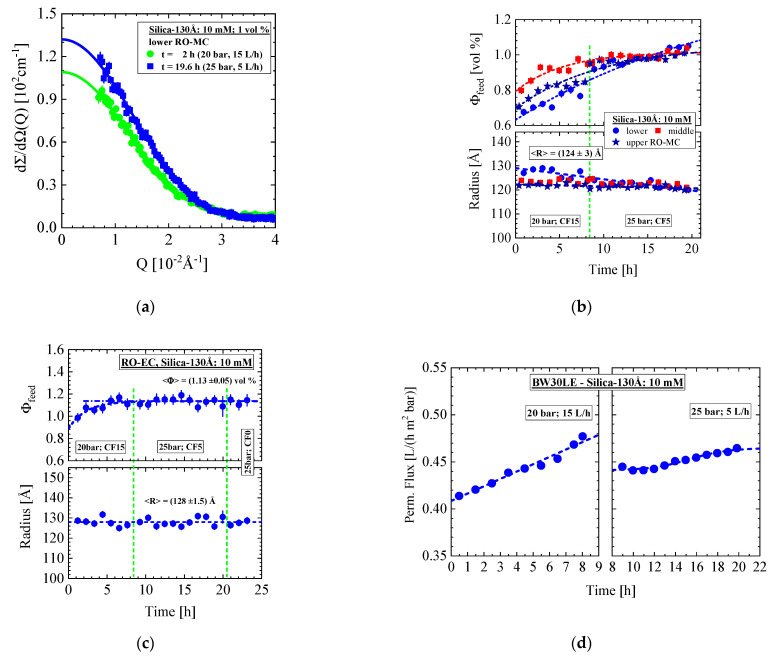
(**a**) Scattered intensity of the Silica-130Å colloids of 1 vol% nominal volume fraction in aqueous dispersion of 10 mM ionic strength. (**b**) The silica volume fraction and radius from all positions of the RO-MC cell versus time and (**c**) the corresponding parameters determined for RO-EC cell including the values from dead-end condition. (**d**) Permeate flux at both conditions.

**Figure 6 membranes-11-00413-f006:**
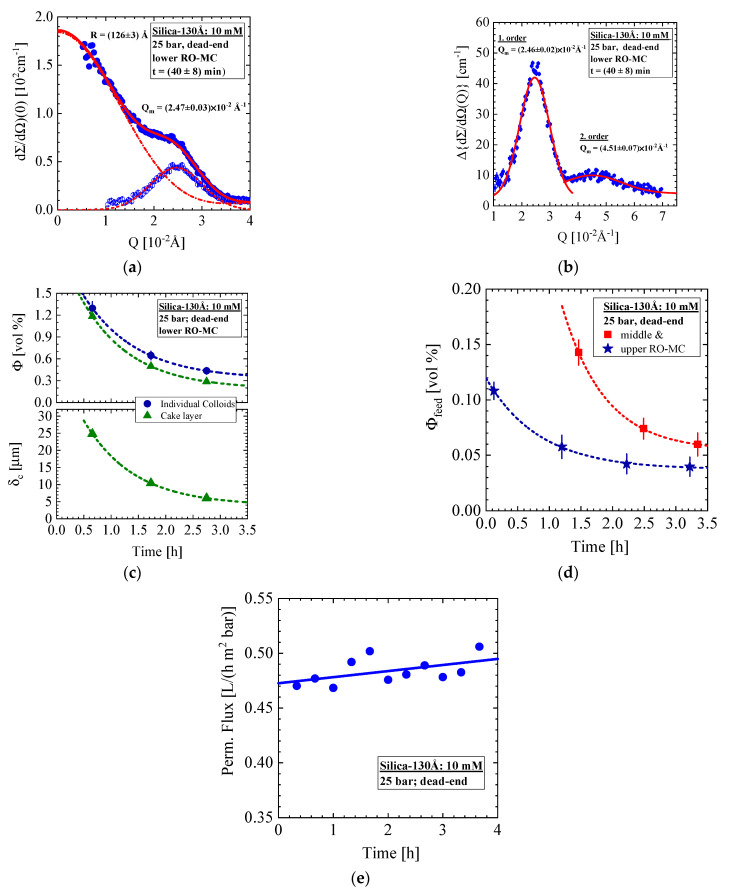
(**a**) Same dispersion but at dead-end and 25 bar condition as in Figure 5. (**a**) Scattering showing after 40 min individual colloids and formation of silica crystal called cake layer. (**b**) Scattering from cake layer after subtraction scattering from individual colloids. (**c**) Declining silica volume fraction of individual colloids and cake layer as well as cake thickness. (**d**) Colloidal feed solutions of middle and upper position of the membrane show an order of magnitude smaller volume fraction. (**e**) Permeate flux of the corresponding experiment.

**Figure 7 membranes-11-00413-f007:**
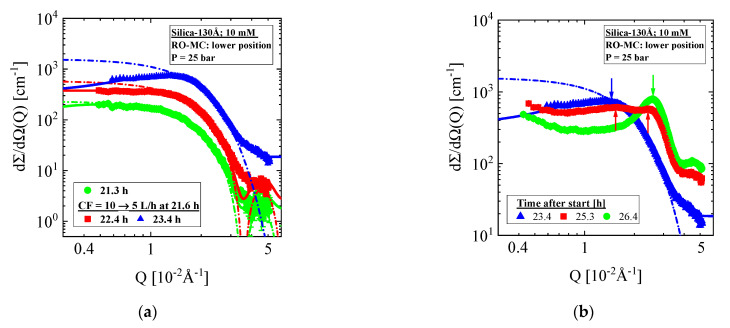
RO-MC cell, lower window: (**a**,**b**) Evolution of scattering patterns when changing the external conditions from CF = 10 L/h to 5 L/h. Strong concentration polarization immediately starts at the lower part of the membrane, in a first step showing interparticle correlation. It takes 4–5 h to form a stable cake layer. (**c**) Volume fraction and radius of disordered silica colloids. (**d**) Amplitude and periodicity of the cake layer.

**Figure 8 membranes-11-00413-f008:**
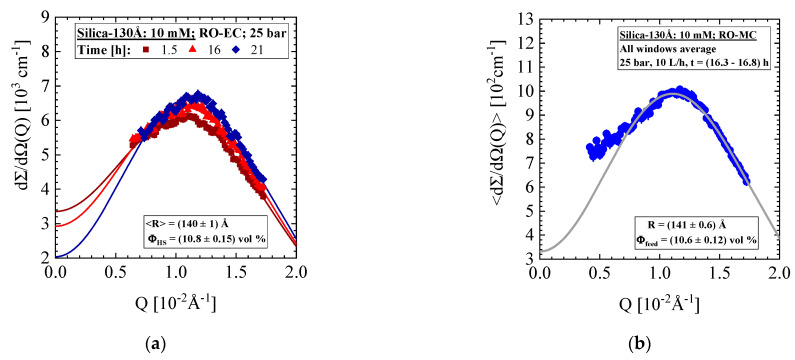
SANS data of nominal 4.74 vol% Silica-130Å dispersion of 10 mM ionic strength. The scattering patterns of the RO-EC cell in (**a**) and RO-MC cell in (**b**) show an interference peak at about Q_m_ = 1.1 × 10^−2^ Å^−1^ due to the larger concentration and repulsive interaction of the colloids at 10 mM. (**c**) Volume fraction of silica from all positions of RO-MC and RO-EC cells with time being relatively insensitive to cross-flow showing about twice as large concentration at the front of the membrane. (**d**) Corresponding permeate flux showing the usual behavior of exponential decline.

**Figure 9 membranes-11-00413-f009:**
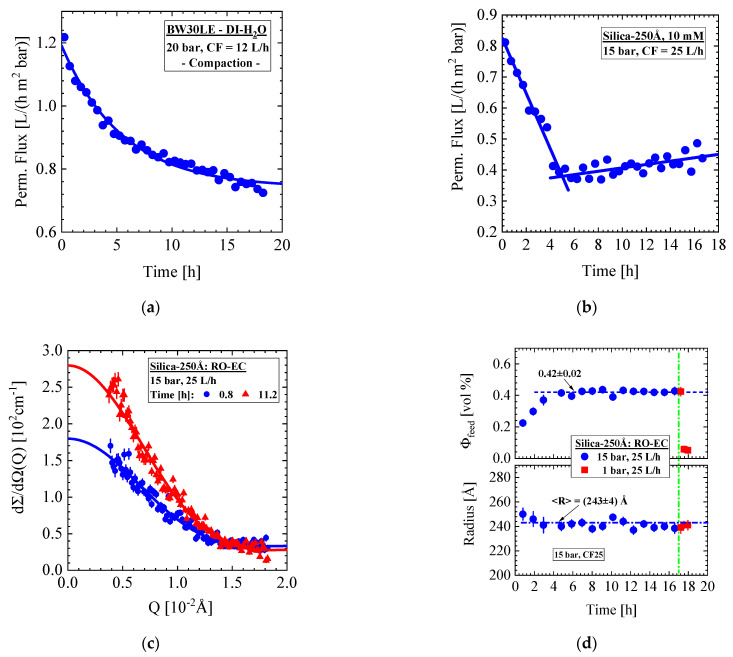
Aqueous dispersion of Silica-250Å of 10 mM ionic strength and a nominal 0.4 vol% volume fraction in the RO-EC cell. (**a**,**b**) show permeate flux during compaction with DI-H_2_O at 25 bar and 12 L/h cross-flow followed after 6 h by the desalination operation. Figure (**c**) shows two scattering curves after about 1 and 12 h of operation. (**d**) Volume fraction and colloid radius versus time. The operation was stopped after 17 h (green dashed-dotted line) for cleaning of the membrane with DI-H_2_O at 25 L/h. Strong decline in silica volume fraction ($\Phi_{\mathrm{feed}}$) is observed.

**Figure 10 membranes-11-00413-f010:**
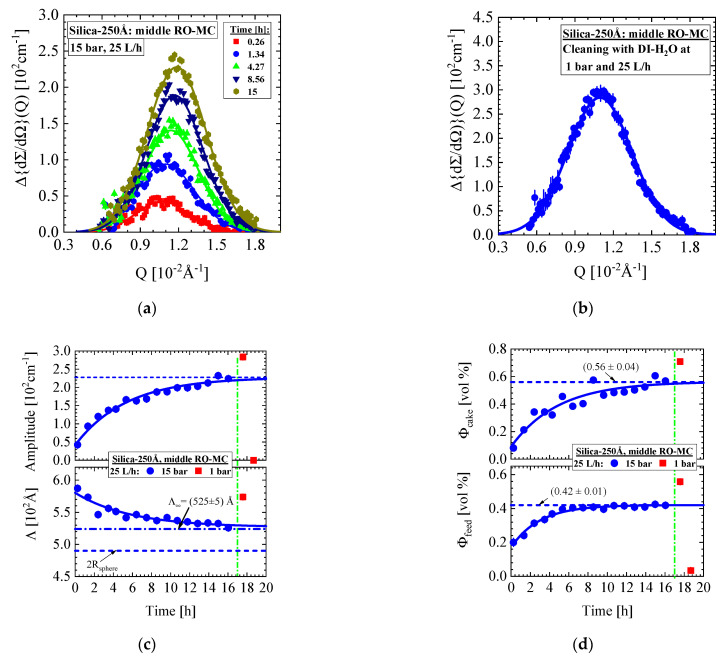
Middle part of the RO-MC cell with same dispersion as in Figure 9. The fits of scattering data were performed with a fixed radius of 245 Å. After 17 h, a cleaning procedure was started with DI-H_2_O at 1 bar and 25 L/h (vertical dashed-dotted line) depicted as red (∎) symbols. (**a**) Bragg scattering shows the evolution of cake layer. (**b**) Scattering from cake layer during in the first stage of cleaning before disappearing. (**c**) Amplitude A and periodicity Λ determined from fitting with Equation (A6). (**d**) Colloid volume fraction of feed and cake normalized to the channel height of the cell. (**e**) Porosity and thickness of the cake layer versus time showing compactification and increase in the cake layer, respectively.

**Figure 11 membranes-11-00413-f011:**
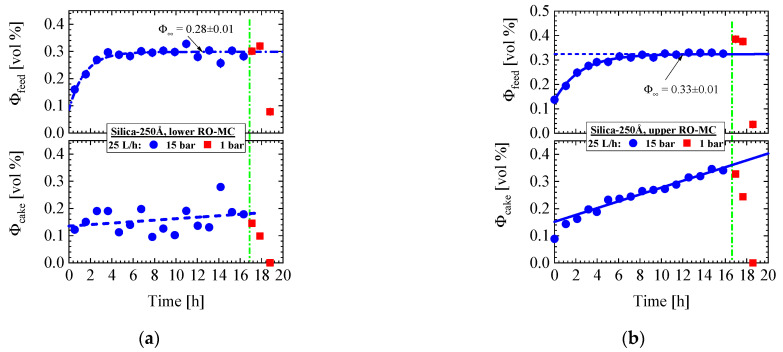
Volume fraction of feed and cake of the lower (**a**) and upper (**b**) part of the RO-MC cell. The values during cleaning procedure are given as red squares.

**Figure 12 membranes-11-00413-f012:**
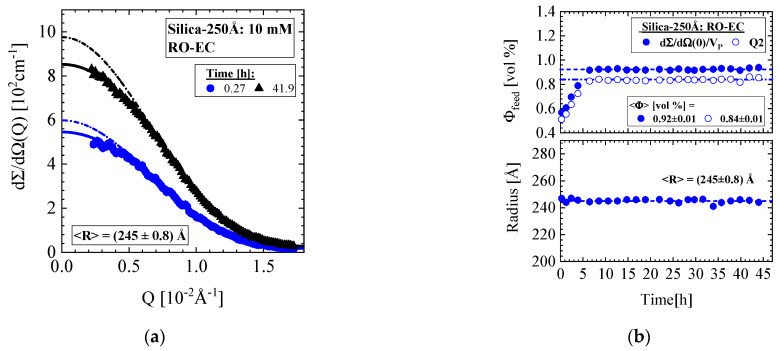
(**a**) Scattering pattern of Silica-250Å colloids of 1.09 vol% nominal concentration determined in the RO-EC cell. (**b**) Silica volume fraction and radius versus time. The volume fraction was determined from ${d\Sigma /d\Omega (0)/V}_{P}$ and Q2 on basis of Equations (A3) and (A7), respectively.

**Figure 13 membranes-11-00413-f013:**
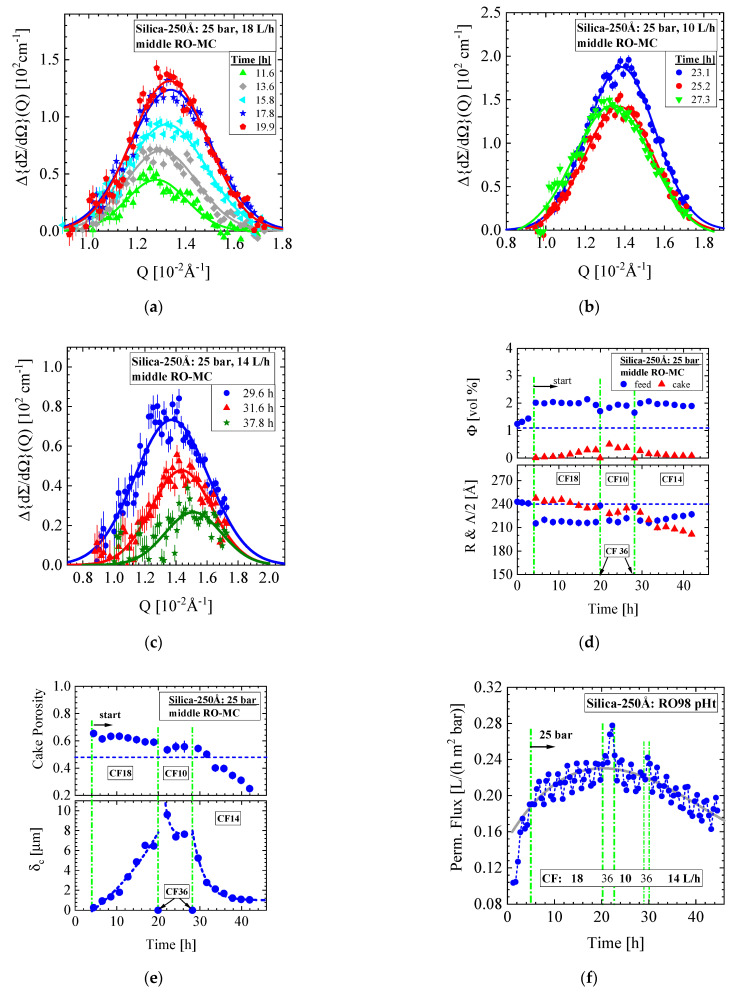
(**a**–**c**) Scattering data of the silica cake formation at the middle part of membrane exposed to 25 bar and cross-flow (CF) of 18, 10, and 14 L/h. Before changing the cross-flow conditions, a CF of 36 L/h was applied for redissolution of the cake layer. (**d**) Volume fraction of disordered and crystallized silica together with the corresponding radius (R) and half periodic distance Λ/2 = π/Q_m_. (**e**) Cake porosity (ε) and thickness (δ_c_) of the cake layer. For the cross-flow of 14 L/h, the porosity becomes smaller than for simple cubic crystals, i.e., ε = 0.48. An increase and decrease in cake layer thickness at, respectively, CF = 18 L/h and the lower ones at 10 and 14 L/h are accompanied by continuous compaction of the cake, which for 14 L/h becomes even larger as for simple cubic crystals. (**f**) Permeate flux during operation.

**Figure 14 membranes-11-00413-f014:**
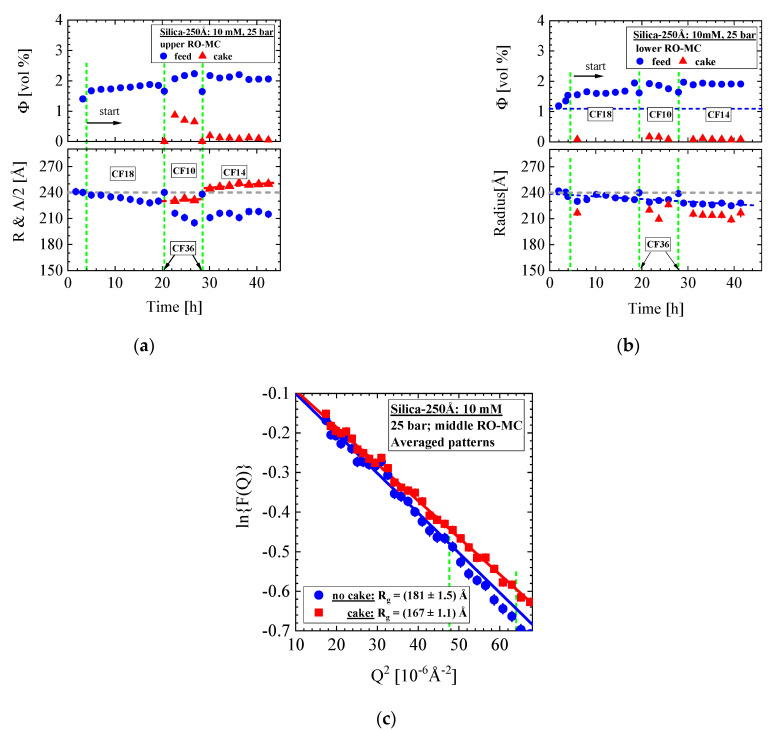
(**a**,**b**) Volume fraction, radius, and periodicity of colloids in front of the membrane of upper and lower part of the RO-MC cell. (**c**) Guinier plot of the formfactor F(Q) (Equation (A2) from averaged scattering curves showing absence (●) and presence (∎) of cake layer. Vertical dashed lines represent the upper limit of fitting.

**Figure 15 membranes-11-00413-f015:**
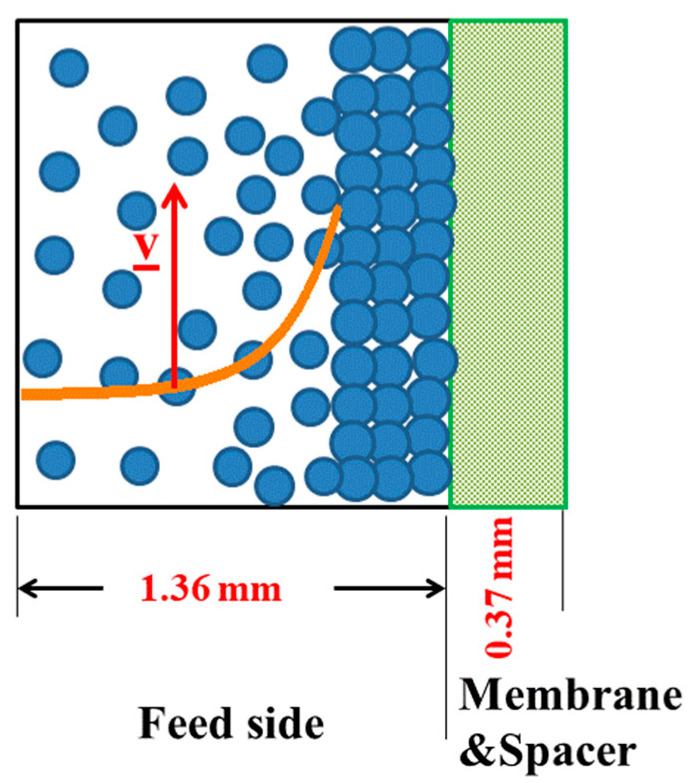
Schematic cross-section of the channel of the RO-MC cell with dimensions of the middle window and brackish water membrane BW30LE at 25 bar showing feed solution and cake layer on top of the membrane surface. The orange curve qualitatively represents the exponential decay of the silica distribution normal to the membrane surface. The cross-flow of 18 L/h corresponds to a cross-flow velocity of v ≅ 8.2 cm/s. The channel area of the feed is 0.61 cm^2^. Upper and lower windows show a 78% smaller channel height, thereby a correspondingly larger cross-flow velocity of v ≅ 10 cm/s (Figure 2).

**Table 1 membranes-11-00413-t001:** Parameters of the silica colloids. The size of the colloids was determined from SANS and TEM according to the data in Figure 1 as well as from DLS. Mass density d_S_ is relevant to determine the coherent scattering length density ρ and their difference Δρ with respect to the solvent water, the last parameter determining the scattering contrast. Coherent scattering length of the monovalent ions ${\mathrm{NH}}_{4}{}^{+}$ (ammonium) and ${\mathrm{Cl}}^{-}$.

Silica Colloid	R (Å) (Silica-130Å)	R (Å) (Silica-250Å)	d_S_ (Amorphous Silica) (g/mL)	ρ (SiO_2_) (10^10^cm^−2^)	Δρ (SiO_2_) (10^10^ cm^−2^)	$b\;({\mathrm{NH}}_{4}{}^{+}$$;{\mathrm{Cl}}^{-})$ (10^12^ cm)
TEM	116 (± 19)	215 (± 24)	2.196 (2.11; [24])	---------	---------	---------
DLS	160	255		---------	---------	---------
SANS	133 ± 1	247 ± 1		3.47	4.03	−0.569; 0.958

**Table 2 membranes-11-00413-t002:** Membranes and silica feed solutions. For all samples, the ionic strength and pH were set by ammonia buffer, i.e., by only the monovalent ions ${\mathrm{NH}}_{4}{}^{+}$ (ammonium cation) and ${\mathrm{Cl}}^{-}$.

Silica Colloid	Silica-130Å			Silica-250Å	
Membrane	RO-BW30LE		RO98 pHt	RO-BW30LE	RO98 pHt
Nominal concentration (vol%)	1	1	4.74	0.4	1.09
Ionic strength (mM)	1.2	10			
pH	9.5	9.6		9.6	
Zeta potential (mV)	−92	−77			

## Data Availability

Exclude this statement.

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
