# Peer review of "Silica Fouling in Reverse Osmosis Systems–Operando Small-Angle Neutron Scattering Studies"

_membranes, 2021, doi:10.3390/membranes11060413_

Round 1

Reviewer 1 Report

The paper entitled “Silica fouling in reverse osmosis systems – Operando small-an-gle neutron scattering studies” and written by Vitaliy Pipich et al. is interesting and reports the problem of silica fouling in brackish water reverse osmosis desalination. This issue has been quite challenging in the industry forcing to reduce flux recovery and dosing antiscalant in desalination plants.My recommendation is a minor revision based on the following comments:

  1. The style of cites and references is not according with the style of the journal.
  2. In the abstract, please, write the meaning of SANS.
  3. The section introduction is quite short; authors should extend it by incorporating information about the silica impact on RO system. I know it is difficult to take into consideration every single work related with the topic of this manuscript but, the authors should include, for example, how the silica concentration limits the flux recovery and increases the cost in brackish water desalination (limiting maximum flux recovery and antiscalant costs), impact of silica fouling in long-term operation in RO systems, etc. Some papers should be included. Here you are some suggestions:
    1. Silica scaling of reverse osmosis membranes preconditioned by natural organic matter
    2. Antiscalant cost and maximum water recovery in reverse osmosis for different inorganic composition of groundwater
    3. Silica treatment technologies in reverse osmosis for industrial desalination: A review
    4. Silica fouling during groundwater RO treatment: the effect of colloids' radius of curvature on dissolution and polymerisation
    5. Estimation of maximum water recovery in RO desalination for different feedwater inorganic compositions
    6. Inorganic scaling in reverse osmosis (RO) desalination: Mechanisms, monitoring, and inhibition strategies
    7. A two-dimensional numerical model for silica colloidal fouling in a spacer-filled reverse osmosis membrane system
    8. Combined silica and sodium alginate fouling of spiral-wound reverse osmosis membranes for seawater desalination
    9. Silica and silicate precipitation as limiting factors in high-recovery reverse osmosis operations
    10. Silica fouling and cleaning of reverse osmosis membranes
  4. Page 2, second line. I understand that the authors had to include reference after a statement to be supported, but, five citations between bracket I think it is too much (not appropriate) please, try to reduce them.
  5. In section 2, reverse osmosis was already abbreviated to RO in section introduction. Please, check the entire document.
  6. The format of the Figures’ captions is not appropriate, font size is not according with the rest of the text and I do not understand why it is framed. Same happens with the Tables.
  7. In Table 1, in its caption the variables are written in a different font and not in italics. Variables should be written in italics. Same happens in page 4.
  8. In page 4, please add the manufacturer of the membranes.
  9. Page 6, section 3, five citations in a row is too much, please, try to reduce them.
  10. I think there is a typo in the end of page 15. Please check the entire document
  11. Was any membrane fouling index (SDI, MFI) measured during the experiment.
  12. How this research helps to improve the operation of real full-scale RO desalination plants?

Author Response

Reviewer 1

The paper entitled “Silica fouling in reverse osmosis systems – Operando small-angle neutron scattering studies” and written by Vitaliy Pipich et al. is interesting and reports the problem of silica fouling in brackish water reverse osmosis desalination. This issue has been quite challenging in the industry forcing to reduce flux recovery and dosing antiscalant in desalination plants. My recommendation is a minor revision based on the following comments:

  1. The style of cites and references is not according with the style of the journal.
  2. This was changed in the transformation of the submitted manuscript to MDPI style by the publisher.
  3. In the abstract, please, write the meaning of SANS.
  4. It is done
  5. The section introduction is quite short; authors should extend it by incorporating information about the silica impact on RO system. I know it is difficult to take into consideration every single work related with the topic of this manuscript but, the authors should include, for example, how the silica concentration limits the flux recovery and increases the cost in brackish water desalination (limiting maximum flux recovery and antiscalant costs), impact of silica fouling in long-term operation in RO systems, etc. Some papers should be included. Here you are some suggestions:
    1. Silica scaling of reverse osmosis membranes preconditioned by natural organic matter
    2. Antiscalant cost and maximum water recovery in reverse osmosis for different inorganic composition of groundwater
    3. Silica treatment technologies in reverse osmosis for industrial desalination: A review
    4. Silica fouling during groundwater RO treatment: the effect of colloids' radius of curvature on dissolution and polymerisation
    5. Estimation of maximum water recovery in RO desalination for different feedwater inorganic compositions
    6. Inorganic scaling in reverse osmosis (RO) desalination: Mechanisms, monitoring, and inhibition strategies
    7. A two-dimensional numerical model for silica colloidal fouling in a spacer-filled reverse osmosis membrane system
    8. Combined silica and sodium alginate fouling of spiral-wound reverse osmosis membranes for seawater desalination
    9. Silica and silicate precipitation as limiting factors in high-recovery reverse osmosis operations
    10. Silica fouling and cleaning of reverse osmosis membranes

We followed the advice of both Reviewers and largely extended the Introduction by “the silica impact on RO system” as well as adding several new references supporting this aspect.

  1. Page 2, second line. I understand that the authors had to include reference after a statement to be supported, but, five citations between bracket I think it is too much (not appropriate) please, try to reduce them.
  2. We erased references 4 and 5, i.e. the classical papers of Derjaguin et al. and Verwey et al., which were cited in respect to the founders of the DLVO theory.
  3. In section 2, reverse osmosis was already abbreviated to RO in section introduction. Please, check the entire document.
  4. The Reviewer is correct. We did so and changed reverse osmosis to RO
  5. The format of the Figures’ captions is not appropriate, font size is not according with the rest of the text and I do not understand why it is framed. Same happens with the Tables.
  6. The style was changed by the publisher.
  7. In Table 1, in its caption the variables are written in a different font and not in italics. Variables should be written in italics. Same happens in page 4.
  8. see answer to 1.
  9. In page 4, please add the manufacturer of the membranes.
  10. We made a note in the text and added the names of the manufacturers in the title of Table A1
  11. Page 6, section 3, five citations in a row is too much, please, try to reduce them.
  12. see answer to 4.
  13. I think there is a typo in the end of page 15. Please check the entire document
  14. Yes, thank you! We corrected the mistakes
  15. Was any membrane fouling index (SDI, MFI) measured during the experiment.
  16. We didn’t measure any fouling index of the feed. The limitations of the SDI and MFI indices are discussed in chapter 3 of ref [1]. In contrast, we directly characterized the fouling behavior of the feeds, i.e. determined the averaged silica volume fraction in front of the membrane (concentration polarization) as well as volume fraction, thickness and porosity of the silica cake. An example is given in Figure A7 showing the evolution of cake resistance during the desalination as evaluated from Equation (A8).    
  17. How this research helps to improve the operation of real full-scale RO desalination plants?
  18. This is a very important and a challenging question! Reviewer 1 should consider that the present study is the first one of its kind. Nevertheless, the results of this paper will improve the understanding of mechanisms of silica scaling in RO processes, that enable better coping with the scaling problems. For example, design of anti-scaling modifications of RO membranes by considering the detailed scaling mechanism observed in the present study. We were very much surprised obtaining such detailed information about silica colloidal fouling and in particular about cake formation. There are results about the reversibility and irreversibility of cake layer formation (Figure 10e) as well the effect of the cake layer on permeate flux (Fig. 9b), which could become relevant for improving the operation of real full-scale RO desalination plants. But these first observations still need further detailed and focused operando SANS experiments.

Reviewer 2 Report

The paper presents the outcome of some investigations exploiting small-angle neutron scattering to analyse the phenomenon of colloidal fouling on two reverse osmosis (RO) membranes under conditions which are stated to be close to the real ones that are observed in implemented RO desalination techniques. This approach shows the advantage of providing a direct insight into the fouling process on microscopic length scale. The investigations were conducted for different aqueous silica dispersions conditions (i.e. different combination of colloidal radius, volume fraction, ionic strength) and permeate flux.

The topic of the paper is interesting and perfectly fits the aims and scope of the journal.

The introductory section presents a quite synthetic and superficial analysis of the state of the art, with many lump references. This aspect needs to be improved, as the authors should clearly explain the relevance and pertinence of each cited paper with respect to the context in which it is cited. What is also missing, is a comprehensive discussion of the problems that fouling might cause in real world application. This is a very relevant aspect, that should be emphasized and deeply discussed in order to highlight the importance and contribution of the proposed investigation. There is a wide variety of literature works e.g. on industrial application of RO. The authors ca see, for instance, the following papers:

  • Turner, C. et al. Silica fouling during groundwater RO treatment: The effect of colloids’ radius of curvature on dissolution and polymerization, (2020) Water Research, 168.
  • Wang, F. et al. Combined Precipitative and Colloidal Fouling of Reverse Osmosis Membranes, (2019) Journal of Environmental Engineering (United States), 145 (8).
  • Qrenawi, L.I., Abuhabib, A.A., A review on sources, types, mechanisms, characteristics, impacts and control strategies of fouling in ro membrane systems, (2020) Desalination and Water Treatment, 208, pp. 43-69.

Section 2 describes the experimental setup, the adopted instrumentation and the experimental procedure. The provided description is overall clear and comprehensive. On the other hand, Section 3, which provides the theoretical background on the stability of aqueous silica dispersions and justifying the selected scenarios, is not very clear and should be more deeply elaborated.

The presentation of the experimental results, which is provided in Section 4, is overall clear and comprehensive and the following discussion of the results is detailed and well supported by the provided data. However, the last section mixes the discussion of the results and the conclusions, and this is a weaknesses. A deep discussion of the results is indeed beneficial, but a conclusions section is needed in order not only to summarize the contents of the paper but, mostly, to highlight the main results, their general applicability and their consequences. This is totally missing in the present version of the paper.

From the formal point of view, any acronym should be explained the first time it is used and the use of acronyms should be avoided in the abstract. Moreover the caption of the figures is not in the requested format. This hold also for the tables and the Appendixes.

Author Response

Reviewer 2

The paper presents the outcome of some investigations exploiting small-angle neutron scattering to analyse the phenomenon of colloidal fouling on two reverse osmosis (RO) membranes under conditions which are stated to be close to the real ones that are observed in implemented RO desalination techniques. This approach shows the advantage of providing a direct insight into the fouling process on microscopic length scale. The investigations were conducted for different aqueous silica dispersions conditions (i.e. different combination of colloidal radius, volume fraction, ionic strength) and permeate flux.

The topic of the paper is interesting and perfectly fits the aims and scope of the journal.

  1. The introductory section presents a quite synthetic and superficial analysis of the state of the art, with many lump references. This aspect needs to be improved, as the authors should clearly explain the relevance and pertinence of each cited paper with respect to the context in which it is cited. What is also missing, is a comprehensive discussion of the problems that fouling might cause in real world application. This is a very relevant aspect, that should be emphasized and deeply discussed in order to highlight the importance and contribution of the proposed investigation. There is a wide variety of literature works e.g. on industrial application of RO. The authors can see, for instance, the following papers:
    1. Turner, C. et al. Silica fouling during groundwater RO treatment: The effect of colloids’ radius of curvature on dissolution and polymerization, (2020) Water Research, 168.
    2. Wang, F. et al. Combined Precipitative and Colloidal Fouling of Reverse Osmosis Membranes, (2019) Journal of Environmental Engineering (United States), 145 (8).
    3. Qrenawi, L.I., Abuhabib, A.A., A review on sources, types, mechanisms, characteristics, impacts and control strategies of fouling in ro membrane systems, (2020) Desalination and Water Treatment, 208, pp. 43-69.

See answer to comment 3 of Reviewer 1.

  1. Section 2 describes the experimental setup, the adopted instrumentation and the experimental procedure. The provided description is overall clear and comprehensive.
  2. Thanks!
  3. On the other hand, Section 3, which provides the theoretical background on the stability of aqueous silica dispersions and justifying the selected scenarios, is not very clear and should be more deeply elaborated.

The DLVO theory is a well-established theory. So, we present a short introduction, giving the relevant equations used in the calculations of interaction potential curves depicted in Figure3. These curves are meant to sketch the stability of the present colloidal dispersions in a semi-quantitative way and show its dependence on the particle size and the salt concentration expressed as the ionic strength. In order to contribute to a better readability of the manuscript, some minor modifications were made in section 3. For further details the readers are referred to one of the two books cited in the manuscript (now ref. 18 Israelachvili or ref. 19 Russel et al.) that treat the subject in much more detail.

  1. The presentation of the experimental results, which is provided in Section 4, is overall clear and comprehensive and the following discussion of the results is detailed and well supported by the provided data. However, the last section mixes the discussion of the results and the conclusions, and this is a weakness. A deep discussion of the results is indeed beneficial, but a conclusions section is needed in order not only to summarize the contents of the paper but, mostly, to highlight the main results, their general applicability and their consequences. This is totally missing in the present version of the paper.

The Reviewer probably means the interpretation of the observed small decline of the colloidal radius in front of the membrane (Figure 14 of section 4.5) given in the last sentences of section 4: “The origin of the decline of radii in presence of cake is unclear to us and only observed for the RO98 pHt membrane. This observation might be linked to the formation of the gel layer of approximately 10 Å thickness at the silica surface when exposed to water and its effect of a larger stability of silica dispersion due to non-DLVO interaction. A decline of colloid radius according to the disappearance of the gel lay would to some extend decline the stability of the silica dispersion.” We erased these sentences and added one sentence to section 5.2 of the Discussion and Interpretation part, where we proposed an interpretation of this observation.

  1. From the formal point of view, any acronym should be explained the first time it is used and the use of acronyms should be avoided in the abstract. Moreover, the caption of the figures is not in the requested format. This hold also for the tables and the Appendixes.
  2. This was caused from transformation of the submitted Word document to MDPI style by the Publisher.

Reviewer 3 Report

The current study introduces an extensive overview on the application of small angle neutron scattering technique for studying silica scaling of RO membranes at different operating conditions.

  • Despite the fact that the current study introduces a comprehensive description of the scientific background, equations and results, the manuscript is quite lengthy and not well organized. Also, it is believed that some results can be provided as supplementary data.
  • The summary of the results provided in section 5 might be better combined to section 4.
  • A separate and brief conclusion (not a summary) of the main scientific outputs (not the technical ones) regarding silica scaling mechanisms and the influences of different operating conditions should be added.
  • The citation of references in the main text should be changed into Arabic numbers not latin numbers.
  • The symbols and units all over the text should be reviewed (there are many "?").
  • What are the reasons for using ammonia/Cl- buffer? and why were the filtration experiments performed at such high pH (9-10). Silica particles and TFC membranes are already negatively charged in the normal pH range (6-7), which is also more realistic.
  • Since the SANS instrument is sensitive to scattering particles between few A up to 100 nm, why not were bigger silica particles (50 nm, or more) tested as well? The employed small silica particles seem to be smaller than the common range known in real water samples.
  • The text in page 5 concerning the measurement of zeta potential is hard to understand. Also, the influence of ionic strength was not properly considered. It is known that increasing the ionic strength might result in low measured zetapotential values because of the compaction of the EDL.
  • It is also not known from the text, how was ionic strength of the feed solution was adjusted?
  • Figures 4,5,7,8: the impact of crossflow velocity is not clear and cannot be understood from the text.
  • It is also not clear in the manuscript why the authors tested two different types of RO membranes. The membranes should be properly characterized, most importantly the salt retention in order to learn about the concentration polarization ability.

Round 2

Reviewer 2 Report

The paper has been revised according to the suggestions and observations provided by the reviewers. It was reworked and drastically improved.

I still think that a concise Section named "Conclusion" is missing at the end of the paper and the format is still far from the classical one. Nonetheless, from the substantial point of view, the paper now is suitable to publication.